# A “Special” Solvent to Prepare Alloyed Pd_2_Ni_1_ Nanoclusters on a MWCNT Catalyst for Enhanced Electrocatalytic Oxidation of Formic Acid

**DOI:** 10.3390/nano13040755

**Published:** 2023-02-17

**Authors:** Pingping Yang, Li Zhang, Xuejiao Wei, Shiming Dong, Wenting Cao, Dong Ma, Yuejun Ouyang, Yixi Xie, Junjie Fei

**Affiliations:** 1College of Chemistry and Materials Engineering, Huaihua University, Huaihua 418008, China; 2College of Chemistry, Xiangtan University, Xiangtan 411105, China

**Keywords:** Pd_2_Ni_1_ nanocluster, doping, Pd–Ni bond, deep eutectic solvents, formic acid oxidation reaction

## Abstract

Herein, an electrocatalyst with Pd_2_Ni_1_ nanoclusters, supporting multiwalled carbon nanotubes (MWCNTs) (referred to Pd_2_Ni_1_/CNTs), was fabricated with deep eutectic solvents (DES), which simultaneously served as reducing agent, dispersant, and solvent. The mass activity of the catalyst for formic acid oxidation reaction (FAOR) was increased nearly four times compared to a Pd/C catalyst. The excellent catalytic activity of Pd_2_Ni_1_/CNTs was ascribed to the special nanocluster structure and appropriate Ni doping, which changed the electron configuration of Pd to reduce the d-band and to produce a Pd–Ni bond as a new active sites. These newly added Ni sites obtained more OH^−^ to release more effective active sites by interacting with the intermediate produced in the first step of FAOR. Hence, this study provides a new method for preparing a Pd–Ni catalyst with high catalytic performance.

## 1. Introduction

Liquid fuel cells are booming related to achieving carbon neutrality [1,2,3,4,5]. Direct formic acid fuel cell (DFAFC) is a kind of important fuel cell and attracts much attention [6,7,8]. However, low conversion efficiency of formic acid has always been a difficult problem. This is partly due to the slow kinetics of multi-electron transfer during the formic acid oxidation reaction (FAOR).

Palladium (Pd) is a kind of precious metal that has a good catalytic effect towards FAOR [9]. Besides, various Pd-based bimetallic alloy catalysts were prepared to improve the FAOR (e.g., PdAg [10], PdSn [11], PdNi [12], PdBi [13], and PdRu [14]). Among these alloy catalysts, PdNi binary catalysts have attracted much attention. Gao et al. [15] prepared a Pd–Ni/C catalyst that shows excellent performance for FAOR. Han et al. [16] fabricated Pd_x_Ni/rGO catalyst showing enhanced mass activity toward FAOR. The doping of the Ni is conducive to change the electronic structure around the Pd atom, which is conducive to improve the toxicity resistance of the catalyst. Although the addition of the second metal has improved the catalytic performance of the single Pd catalyst for formic acid to a certain extent, in order to meet practical application, the catalyst performance needs to be improved.

Optimizing the morphology of Pd binary catalysts is helpful to improve their catalytic performance. Among all kinds of heterogeneous structures, catalysts with nanocluster structures have attracted much attention. This structure generally has rough surfaces and abundant low-coordination atoms, leading to larger electrochemically active surface area (ECSA) and improved electrocatalytic performance [17]. Besides, PdNi alloy nanocluster catalysts have not yet been reported so far.

For precious metals, doping can improve their catalytic activity [18]. However, too large or too small of a doping ratio cannot obtain the best catalytic performance. Hence, it is of great significance to explore the appropriate doping ratio to obtain a catalyst with high performance.

Recently, carbon materials have been widely used as carriers due to their excellent electron transport ability [19]. Among these carbon materials, MWCNTs have attracted much attention in fuel cells [20]. Hence, in this study, we introduced MWCNTs as carriers to improve the electron transport capacity of Pd_2_Ni_1_ nanoclusters.

It is well known that smaller nanoparticles can increase ECSA for more effective active sites. Deep eutectic solvents (DES) are an analogue of ionic liquids, with many advantages, wide electrochemical window, high electroconductivity, non-toxicity, thermal stability, biodegradability, and negligible vapor pressure, in addition to other advantages [21]. Our group [19,22,23,24,25] has also concluded that this green solvent can effectively reduce the size of nanoparticles in previous research. In addition, some research groups have successfully prepared PdSn [26] and PdAg [27] catalysts in DES. Specially, Hammons research group [28] used DES to stabilize the resulting Pd nanoparticles. Therefore, DES is a potential solvent for fabrication of catalysts with excellent performance.

The highlight of this work is that the Pd_2_Ni_1_-CNTs catalyst was prepared in DES for the first time. In this study, porous Pd_2_Ni_1_ nanoclusters were obtained in DES, so as to obtain larger ESCA to provide more active sites and thus obtain greater catalytic activity. In addition, conventional PdNi-CNT catalysts are usually prepared in a water system or in toxic organic solvents, and additional reducing agents are needed. Besides, DES is a green and environmentally friendly solvent used in this study. It serves as a reducing agent, dispersant, and solvent simultaneously in the preparation of a catalyst. In this study, Pd_2_Ni_1_ alloy nanoclusters on a MWCNT catalyst was fabricated with a DES-assisted environment (Figure 1). Subsequent electrochemical tests showed that the catalyst had good CO resistance and excellent catalytic activity for formic acid.

## 2. Results and Discussion

Figure 1a shows the XRD patterns of Pd_2_Ni_1_/CNTs and Pd/CNTs. The diffraction peak near 26° is caused by C(002) of the carrier [19,23]. Compared to the Pd diffraction peaks of Pd/CNTs (39.2°, 44.8°, 66.5°, and 79.8°), the Pd peaks of Pd_2_Ni_1_/CNTs shifted to the higher 2θ values (39.5°, 45.3°, 66.6°, and 80.3°), suggesting the formation of alloy [29]. This result may be due to the fact that the radius of Ni is smaller than Pd, which compresses the lattice constant of Pd during doping [30]. This phenomenon is also strong evidence for the formation of alloys. Using Scherrer’s equation [31] to calculate the average nanoparticles’ sizes of Pd_2_Ni_1_/CNTs (3.6 nm) (Figure 1b) and Pd/CNTs (4.1 nm) (Appendix A), the small size of nanoparticles in Pd_2_Ni_1_/CNTs resulted in more effective catalytic active sites on the catalyst, which was also the main reason for its large ESCA in electrochemical tests.

Figure 2a–c show the TEM and HRTEM images of Pd_2_Ni_1_/CNTs. The Pd_2_Ni_1_ nanoclusters supported on MWCNTs are clear, and the spacing of Pd(111) crystal planes in Pd_2_Ni_1_/CNTs is 0.219 nm, while the value is (0.222 nm) in Pd/CNTs (Appendix A), providing further evidence of alloying [23]. It is clear that the nanoclusters are made up of smaller nanoparticles, with gaps between them that allow formic acid molecules to be transported through them and oxidized [32]. Figure 2d–g display the HAADF-STEM images of Pd and Ni, which are supported by the EDX line-profiles and spot scanning (Appendix A). It is clear that the Pd and Ni atoms appear in almost the same position. This grounding proves the presence of the alloy, and the cluster structure is also evident.

As shown in Figure 3a, the signal peaks corresponding to C1s, O1s, Pd3d, and Ni2p appear in the full spectrum of Pd_2_Ni_1_/CNTs. This indicates that the Pd_2_Ni_1_/CNTs catalyst has been successfully prepared. As can be seen from Figure 3b, the two C–C and C–O peaks are C1 peaks of MWCNTs-AO. After acidification, the surface of the originally smooth CNTs became rough and defective. Figure 3c shows the peaks of Ni2p (Ni 2p1/2 and Ni 2p3/2). The Ni 2p3/2 spectra show the peaks at 856.0 and 860.8 eV, while the Ni 2p1/2 spectra show the peaks at 872.7 and 874.8 eV, accompanied with satellite peaks at 864.0 and 880.4 eV [33]. The presence of Ni^0^ and Ni^+2^ is attributed to their alloy formation with Pd and further oxidation by exposure to air, respectively. As can be seen from Figure 3d, the peak position of Pd^0^ varied from Pd/CNTs to Pd_2_Ni_1_/CNTs by 0.2 eV (Appendix A), indicating a strong charge transfer interaction between Ni and Pd [34]. The optimized Ni doping can effectively regulate the arrangement of the electron configuration of Pd and inhibit the CO_ads_ toxicity at active sites. This is also one of the important reasons for the improvement of performance.

As shown in Figure 4a, the ECSAs of Pd_2_Ni_1_/CNTs, Pd/CNTs, and Pd/C were calculated to be 53.30 m^2^ g^−1^, 42.70 m^2^ g^−1^, and 40.32 m^2^ g^−1^, which are most likely due to the Pd_2_Ni_1_ structural advantages. Larger ECSA is one of the reasons for the excellent activity of Pd_2_Ni_1_/CNTs, but it is not the main one. In order to obtain more suitable Ni doping, we optimized the atomic ratio of Pd to Ni (Figure 4b). The FAOR current densities of Pd_2_Ni_1_/CNTs, Pd_1_Ni_1_/CNTs, Pd_1_Ni_2_/CNTs, Pd/CNTs, and Pd/C are 3351.6 mA mg_Pd_^−1^, 1832.1 mA mg_Pd_^−1^, 1023.5 mA mg_Pd_^−1^, 2399.7 mA mg_Pd_^−1^, and 810.5 mA mg_Pd_^−1^ (Figure 4b,c). Obviously, Pd_2_Ni_1_/CNTs shows the highest catalytic activity. This phenomenon shows that appropriate Ni doping is necessary, and optimized Ni doping can effectively regulate the exonuclear electron configuration of Pd and thus affect the activity of catalyst. We performed a two-hour stability test on these catalysts (Figure 4d). All curves exhibit significant current attenuation at the initial stage, which was attributed to the formation of toxic species [35]. Finally, the Pd_2_Ni_1_/CNTs maintained excellent performance: 167.6 mA mg_Pt_^−1^ is almost 4.5 and 19.5 times greater compared to the values of Pd/CNTs, 37.2 mA mg_Pt_^−1^ and 8.6 mA mg_Pt_^−1^ for Pd/C, respectively. This phenomenon shows that Ni doping successfully inhibits the aggregation and shedding of active components in the catalyst. Good corrosion resistance of acidified MWCNTs was also demonstrated. These results above illustrate that Pd_2_Ni_1_/CNTs exhibits higher electrocatalytic activity and stability. Additionally, the Pd_2_Ni_1_/CNTs catalyst presents superior FAO mass activity in comparison to the Pd-based catalysts investigated in recent studies, as shown in Appendix A. After CA investigations, the size of nanoparticles in Pd_2_Ni_1_/CNTs is still much smaller (3.8 nm) than that in Pd/CNTs (6.8 nm), as shown in Appendix A. This again demonstrates that the Pd_2_Ni_1_/CNTs catalyst exhibits excellent stability towards FAOR in the acidic environments. This phenomenon is attributed to the fact that the addition of Ni can change the electron configuration of Pd to produce a Pd–Ni bond to prevent the agglomeration of Pd in the electrocatalytic process. In addition, the surface of MWCNT-coated Pd–Ni nanoparticles contains more active sites, which helps to reduce the surface energy of Pd deposition and makes Pd more evenly dispersed.

CO stripping experiments were conducted to test the anti-toxicity of these catalysts (Figure 5). Compared with Pd/CNTs (0.69V) and Pd/C (0.74V), the initial potential of CO oxidation adsorbed by Pd_2_Ni_1_/CNTs moved negatively to 0.66V, indicating that Pd_2_Ni_1_/CNTs has excellent CO oxidation ability, which is attributed to Ni doping. These newly added Ni sites can obtain more oxygen-containing groups (OH^−^) from H_2_O and can release more effectively at active sites to improve CO tolerance [35]. This is also the main reason that the catalyst activity has been greatly improved.

High catalytic activity and robust stability are the keys of DFAFC catalysts. Objectively, the catalytic activity and stability of the Pd_2_Ni_1_/CNTs catalyst prepared by us can only be described as relatively modest. Here are some possible explanations for the result. (1) Carrier CNTs were corroded in an acidic environment, resulting in separation of carrier and catalytic active component alloy NPs. (2) During the operation of the battery, the active components on the carrier may agglomerate or fall off, which may also be one of the reasons for the reduced activity and stability of the catalyst. (3) Catalyst poisoning caused by intermediates (such as CO_ads_) produced during the reaction will also lead to degradation of catalyst performance and stability.

There are several methods to improve the activity of catalysts: (1) modification of carbon materials to improve their corrosion resistance, including non-covalent bonding, doping, or repairing defective carbon materials; (2) too add another metal to form a high entropy alloy helps to mitigate the toxicity (CO_ads_). Therefore, the activity of the Pd_2_Ni_1_/CNTs catalyst still needs to be improved for better application in DFAFC.

## 3. Experimental

### 3.1. Materials

DES (choline chloride/oxalic acid) was prepared by referring to the literature [36]. MWCNT (OD: 10–20 nm, length: ~50 mm, purity > 95 wt%) was purchased from Nanjing Xianfeng Nanomaterial Technology Co., Ltd. Nafion solution (5 wt%) was purchased from Sigma-Aldrich. Choline chloride, oxalic acid, ethanol, PdCl_2_, Ni(NO_3_)_2_, H_2_SO_4_, and HNO_3_ were purchased from Shanghai Chemical Reagent Co., Ltd. (Shanghai, China), and all of the reagents were analytically pure.

### 3.2. Preparation of Catalysts

In general, a mixture of H_2_SO_4_ (60 mL) and HNO_3_ (20 mL), with a volume ratio of 3 to 1, was used to acidify and functionalize MWCNT (1.0 g) (MWCNTs-AO) [37]. An amount of 20 mg MWCNTs-AO, 1.32 mL PdCl_2_/DES solution (5 mg/mL), and 0.44 mL Ni(NO_3_)_2_/DES solution (5 mg/mL) were added to 10 mL DES. Then, this mixture was stirred for three hours at room temperature. The obtained product was washed, with suction filtration, and vacuous dried at 60 °C (referring to Pd_2_Ni_1_/CNTs). In the control experiment, Pd_1_Ni_1_/CNTs, Pd_1_Ni_2_/CNTs, and Pd/CNTs were fabricated. All of the solutions were prepared using DES.

### 3.3. Physical Characterization

The X-ray diffraction (XRD) patterns were obtained using a X-ray diffractometer (Rigaku D/MAX 2500 v/pc, Japan), with Cu and K as radiation sources (l ¼ 1.5406 Å). The X-ray photoelectron spectroscopy (XPS) measurements were carried out using a Physical Electronics PHI Quantum 2000 system, with Al and K as radiation sources, and all of the XPS spectra were calibrated with the C1s line at 284.5 eV. The surface morphologies and microstructures of the prepared catalysts were analyzed using a high-resolution transmission electron microscope (HRTEM, JEOL JEM-2100) with an accelerating voltage of 200 kV. An inductively coupled plasma–optical emission spectrophotometer (ICP-OES, Thermo Electron IRIS Intrepid II XSP, USA) was used to characterize the morphology and content of these catalysts. The Pd contents in Pd/C (20%), Pd/CNTs (16.3%), Pd_1_Ni_1_/CNTs (19.1%), Pd_1_Ni_2_/CNTs (18.7%), and Pd_2_Ni_1_/CNTs (18.6%) were measured (Appendix A).

### 3.4. Electrochemical Measurements

The traditional three-electrode system of the electrochemical workstation was used for all electrochemical measurements. Platinum foil and a saturated calomel electrode (SCE) were used as the counter and reference electrodes, respectively. The working electrode was modified as follows: glass carbon electrode (GC, Φ = 5 mm), 0.05 µm, 0.3 µm, and 1.0 µm alumina powder for polishing and processing, Then, the GC was cleaned with water and ethanol. A 2 mg catalyst was dispersed in a 1 mL solution, including 950 µL redistilled water and 50 μL Nafion solution (0.5 wt%). A 10 μL suspension was dripped over the GC electrode at room temperature. The Pd loading of Pd/C, Pd/CNTs, Pd_1_Ni_1_/CNTs, Pd_1_Ni_2_/CNTs, and Pd_2_Ni_1_/CNTs were 24.6, 24.3, 25.3, 24.6, and 25.1 μg cm^−2^.

These electrocatalytic behaviors were examined in 0.5 M H_2_SO_4_ and 1.0 M HCOOH + 0.5 M H_2_SO_4_ solution. For the CO anti-toxicity test, the CO was first bubbled in H_2_SO_4_ solution for 20 min at −0.2–0.0 V voltage scanning to make the working electrode completely poisoned. N_2_ was then bubbled through the solution for 30 min to remove the CO in the solution. Keep N_2_ above the solution at all times to prevent air interference.

## 4. Conclusions

Herein, we used DES as reducing agent, dispersant, and solvent to obtain Pd_2_Ni_1_/CNTs catalyst, which shows excellent electrocatalytic performance and excellent anti-CO ability compared to Pd_1_Ni_1_/CNTs, Pd_1_Ni_2_/CNTs, Pd/CNTs, and Pd/C. The result is attributed to the virtue of compositional and structural advantages, which are benefits from the special DES. The special nanocluster structures of the Pd_2_Ni_1_ alloy lead to larger ESCA, and optimized Ni doping can effectively regulate the arrangement of the electron configuration of Pd, which effectively inhibits the adsorption of CO_ads_ on the active site. More OH^−^ obtained from Ni atoms can further interact with toxic intermediates to release more effective active sites. This study provides a new way (DES-assisted method) to fabricate Pd–Ni catalysts for fuel cells.

## Data Availability

Please contact Yixi Xie, xieyixige@xtu.edu.cn.

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
