# Peer review of "A “Special” Solvent to Prepare Alloyed Pd2Ni1 Nanoclusters on a MWCNT Catalyst for Enhanced Electrocatalytic Oxidation of Formic Acid"

_nanomaterials, 2023, doi:10.3390/nano13040755_

Round 1

Reviewer 1 Report

1/Lines 33-35: Gao et al. and Han et al. are not included in the References. So,  the notation of the references should be immediately after their mention, e.g. Gao et al. [15]

2/ Line 56: What does "... vapor pressure et al[21]" mean?

3/ Line 59: Put the reference number immediately after the author, e.g Hammons research group [28]

4/Lines 67-68: The materials and experiments section goes after the results

5/ Include a space after the caption of the figures

Reviewer 2 Report

Comments and suggestions are reported in the attached file.

Reviewer 3 Report

The Authors proposed a very interesting topic for research. The paper is publishable, but a revision is required before the manuscript acceptance.

·At the end of the introduction, you should write what is new in the described work, how it differs from the previous ones, etc. Research in this direction is already underway.

·Please enter the origin of all devices and materials. The chapter Experimental requires a more detailed study. What reagents were used, what concentrations, etc.

· What analytical methods for the measurements were used, in relation to the applicable standards? If so, please provide a note about it.

·How many repetitions have been done?

·There is no discussion of the obtained results with other studies.

·Please carefully follow the text of the article, lines 56-56, 76 need editing.

·Line 187-188 - describe what, refer to other studies.

Reviewer 4 Report

This paper by Yang et al. prepared the Pd2Ni1 nanoclusters in deep eutectic solvent on multi-walled carbon nanotubes. The obtained Pd2Ni1 was used for the electrochemical formic acid oxidation. The results are fair, and it is not novel work. I found many mistakes in interpretation, thus, the following issues need to be resolved before publication in this journal.

1.      In the introduction, section author claimed MWCNTs as a carrier to improve electron transport like “we introduced MWCNTs as carrier to improve the electron transport capacity of Pd2Ni1 nanoclusters.” But page 6 claim corrosion resistance. It seems confusing to understand.

2.      Page 3. Authors claimed “Pd2Ni1/CNT diffraction peaks shifted to the higher 2θ values (39.5°, 45.3°, 66.6°, 80.3°)” but in the XRD data seems like a shift in low angle. This is in contrast to the discussion.

3.      Page 3. Provide a reference for the sentence “It is clear that the nanoclusters are made up of smaller nanoparticles with gaps between them that allow formic acid molecules to be transported through them and oxidized.”

4.      Page 4. XPS spectra need to refit with fixed FWHM and the axis must be high energy to low energy. Ni 2p region spectrum fitting is not clear, I cannot see the 2p1/2 spin-orbital splitting.

5.      Page 5. The stability of the current-time curves is difficult to understand. The i-t curve is not supportive of the high stability of Pd2Ni1/CNTs. This should be explained with supporting data like before and after formic acid oxidation like electron microscope or XRD or XPS. 

Round 2

Reviewer 2 Report

The manuscript can be published.

Author Response

No review comments

Reviewer 3 Report

I recommend the article for publication.

Author Response

No review comments

Reviewer 4 Report

The authors revised almost all comments well and improved some.

But I am very sorry about the author's comment on my previous comment, number 4. I am not convinced of the labeling of fitting and oxidation states in Figure 3(c and d). I recommend understanding the fitting rule and identifying the proper oxidation states of Ni and Pd.
